

# Role of miR-10b-5p in the prognosis of breast cancer

Junmin Wang, Yanyun Yan, Zhiqi Zhang and Yali Li

College of Life Sciences, Hunan Normal University, Changsha, Hunan, China

## ABSTRACT

Breast cancer is the leading cause of cancer-related death in women worldwide. Aberrant expression levels of miR-10b-5p in breast cancer has been reported while the molecular mechanism of miR-10b-5p in tumorigenesis remains elusive. Therefore, this study was aimed to investigate the role of miR-10b-5p in breast cancer and the network of its target genes using bioinformatics analysis. In this study, the expression profiles and prognostic value of miR-10b-5p in breast cancer were analyzed from public databases. Association between miR-10b-5p and clinicopathological parameters were analyzed by non-parametric test. Moreover, the optimal target genes of miR-10b-5p were obtained and their expression patterns were examined using starBase and HPA database. Additionally, the role of these target genes in cancer development were explored via Cancer Hallmarks Analytics Tool (CHAT). The protein–protein interaction (PPI) networks were constructed to further investigate the interactive relationships among these genes. Furthermore, GO, KEGG pathway and Reactome pathway analyses were carried out to decipher functions of these target genes. Results demonstrated that miR-10b-5p was down-regulated in breast cancer and low expression of miR-10b-5p was significantly correlated to worse outcome. Five genes, BIRC5, E2F2, KIF2C, FOXM1, and MCM5, were considered as potential key target genes of miR-10b-5p. As expected, higher expression levels of these genes were observed in breast cancer tissues than in normal tissues. Moreover, analysis from CHAT revealed that these genes were mainly involved in sustaining proliferative signaling in cancer development. In addition, PPI networks analysis revealed strong interactions between target genes. GO, KEGG, and Reactome pathway analysis suggested that these target genes of miR-10b-5p in breast cancer were significantly involved in cell cycle. Predicted target genes were further validated by qRT-PCR analysis in human breast cancer cell line MDA-MB-231 transfected with miR-10b mimic or antisense inhibitors. Taken together, our data suggest that miR-10b-5p functions to impede breast carcinoma progression via regulation of its key target genes and hopefully serves as a potential diagnostic and prognostic marker for breast cancer.

Corresponding author
Yali Li, yalili@hunnu.edu.cn

## INTRODUCTION

Breast carcinoma is the most common malignancy and the leading cause of cancer death in females worldwide. It consists of multiple subtypes with distinct morphologies and clinical implications (*Dai et al., 2015*). Traditional classification systems of breast cancer are based

on various biological characteristics, including histological grade, tumor size, lymph node involvement, estrogen receptors (ER), progesterone receptors (PGR), and human epidermal growth factor receptor 2 (HER2) status. With the development of microarrays, a new paradigm in deciphering breast cancer heterogeneity has been developed (*Cooper, 2001*; *Dai et al., 2015*). Using different gene panels, breast tumors were classified into five intrinsic molecular subtypes, i.e., luminal A, luminal B, HER2-enriched, basal and normal-like tumors (*Hu et al., 2006*; *Dowsett et al., 2013*). Although the accuracy of disease prognosis has been increased by emergence of novel subtypes, breast cancer continues to emerge as a major health issue for women due to high incidence and mortality rates. Therefore, novel targets that can be utilized to predict or treat breast cancers are urgently called for.

MicroRNA (miRNA), small non-coding RNA molecules with 19–24 nucleotides in length, are involved in post-transcriptional gene silencing by targeting the 3′ untranslated region (UTR) of target genes (*Ross & Davis, 2012*). It has been demonstrated that miRNAs play crucial roles in almost every biological process, including cell growth, cell cycle regulation, cell differentiation, apoptosis, inflammation, and stress response (*Iorio et al., 2011*). Moreover, miRNA have been associated with tumorigenesis by acting as tumor suppressors or oncogenes (*Kent & Mendell, 2006*), and have been shown to affect multiple hallmarks of cancer, such as sustaining proliferative signaling, evading growth suppressors, and resisting cell death (*Ruan, Fang & Ouyang, 2009*). Thus, disturbances of miRNA expression appear to contribute to tumor initiation, maintenance, and progression, as well as to invasion and metastasis (*Gandellini et al., 2011*).

As a member of the miRNA family, hsa-miR-10b (hsa-miR-10b-5p) was reported to be associated with the oncogenesis of breast cancer. However, current findings regarding the role of miR-10b in breast cancer are controversial. MiR-10b was originally reported to be downregulated in primary breast tumors compared with normal breast tissue (*Iorio et al., 2005*). In a later study, *Ma, Teruya-Feldstein & Weinberg (2007)* found that miR-10b was highly expressed in metastatic breast cancer cells and positively regulated cell migration and invasion while *Gee et al. (2008)* reported opposite observations. *Nassar et al. (2014)* found that notable downregulation of miR-10b was observed in tumor tissues as compared to normal breast tissues, and it can be used as biomarker for early breast cancer detection in the Lebanese population. Moreover, *Meerson et al. (2019)* reported that obesity exacerbated the decrease in miR-10b expression in primary tumors compared to normal tissue, suggesting that the metabolic state may alter the molecular makeup of a tumor. *Singh et al. (2014)* revealed that exosome-mediated transfer of miR-10b significantly promoted cell invasion in breast cancer. Increased miR-10b levels have been described in metastatic breast cancer (*Roth et al., 2010*), while the link between miR-10b and metastasis remains controversial, which may be partly due to the heterogeneity of miR-10b expression in circulating tumor cells (*Gasch et al., 2015*). Collectively, these observations suggest that further studies need to be carried out before drawing conclusions about the function of miR-10b in breast cancer. In this study, we analyzed the expression data of this miRNA and its molecular network of target genes using several online databases to elucidate the potential mechanisms underlying the role of miR-10b in breast cancer.

## MATERIALS AND METHODS

### Expression profile of miR-10b-5p in breast cancer

The Cancer Genome Atlas (TCGA) is a landmark cancer genomics program, providing a large amount of genomic, epigenomic, transcriptomic, and proteomic data (https://www.cancer.gov/tcga). In this study, we obtained the miR-10b-5p expression profile of various human cancer types from a TCGA data online analysis tool (http://bioinfo.life.hust.edu.cn/miR_path/index.html).

starBase is an open-source platform for studying the miRNA-ncRNA, miRNA-mRNA, RNA-RNA, and RBP-mRNA interactions from CLIP-seq, degradome-seq, and RNA-RNA interactome data (*Li et al., 2014*). Here, we analyzed the expression level of miR-10b-5p in breast cancer and adjacent normal tissues using starBase v3.0 project (http://starbase.sysu.edu.cn).

### Prognostic value of miR-10b-5p in breast cancer

The Kaplan–Meier Plotter Database (KMPD), a web-tool established using gene expression data and survival information downloaded from the Gene Expression Omnibus (GEO), is designed to validate survival-associated miRNAs in various cancer types (http://kmplot.com/analysis/) (*Nagy et al., 2018*). In the current study, we used this online tool to confirm the prognostic value of miR-10b-5p in four public databases (METABRIC, TCGA, GSE19783 and GSE40267) (*Lánczky et al., 2016*). KM survival curves, hazard ratio (HR), 95% confidence intervals (CI) and log rank P were obtained to analyze the correlation of miR-10b-5p to the overall survival (OS) in breast cancer. *P* value of <0.05 was considered statistically significant.

### Association between miR-10b-5p and clinical features

LinkedOmics is a publicly available portal (http://linkedomics.org/) that includes multi-omics data from 32 TCGA cancer types (*Vasaikar et al., 2018*). In the present study, we applied LinkedOmics to identify the relationship between miR-10b-5p and clinical features, including PAM50 subtypes, ER. status, PR. status, HER2. status, histological type, race, radiation therapy, tumor purity, and pathologic TNM stage. The differences were analyzed by non-parametric test.

### Target genes prediction and identification

Negatively correlated significant genes of miR-10b-5p in breast cancer were selected using LinkedOmics. Target genes of miR-10b-5p were predicted using starBase v3.0 database, which contains seven bioinformatic algorithms: PITA, RNA22, miRmap, microT, miRanda, PicTar, and Targetscan. Overlapped genes from both LinkedOmics and starBase database were considered as the optimal target genes of miR-10b-5p. Finally, the expression patterns of these genes in breast cancer and normal tissues were compared using starBase v3.0 and the Human Protein Atlas (HPA) database v18.1 (http://www.proteinatlas.org) (*Uhlen et al., 2017*).

## Functional and network analysis of the overlapping target genes

The role of the target genes of miR-10b-5p in cancer development were explored via Cancer Hallmarks Analytics Tool (CHAT) (*Baker et al., 2017*). Subsequently, the protein–protein interaction (PPI) networks were constructed to investigate the interactive relationships among these genes, using STRING database v11.0 (*Szklarczyk et al., 2019*). Gene Ontology (GO), Kyoto Encyclopedia of Genes and Genomes (KEGG) pathway and Reactome pathway analyses were carried out, and enriched GO terms and pathways were identified according to the cut-off value of false discovery rate (FDR) <0.05.

## Quantitative RT-PCR analysis of target genes in MDA-MB-231 cells

MDA-MB-231 cells (obtained from ATCC and preserved in our lab) were seeded in 24-well plate ($1\times10^5$ cells/well) in DMEM (Gibco, Waltham, MA, USA) supplemented with 10% fetal bovine serum (Gibco, Waltham, MA, USA) and 1% penicillin-streptomycin in a humidified incubator at 37 °C with 5% $CO_2$. Cells were transfected with negative control (NC) or miR-10b mimic (50 nM) or miR-10b antisense inhibitors (100 nM; Ribo-bio, Guangzhou, China) using lipofectamine 2000 (Invitrogen, Carlsbad, CA, USA) and Opti-MEM Reduced Serum Medium (Gibco, USA) according to the manufacturer's instructions. After 24 h of incubation, cells were harvested with TRIzol reagent (Invitrogen). PrimeScript RT reagent Kit (Takara) was used to prepare cDNA from total RNA. qRT-PCR was performed using Luminaris Color HiGreen qPCR Master Mix (Thermo Scientific, Waltham, MA, USA) for targets genes.

# RESULTS

## Identification the aberrant expression of miR-10b-5p in breast cancer

As shown in Fig. 1, the expression profile of miR-10b-5p demonstrated that it was down-regulated in most human cancers, such as breast cancer (BRCA), kidney renal papillary cell carcinoma (KIRP), and uterine corpus endometrial carcinoma (UCEC). We next analyzed miR-10b-5p expression using starBase (based on 1085 breast cancer samples and 104 normal samples) and we found that miR-10b-5p level was significantly lower in breast cancer tissues than in adjacent normal tissues ($P = 3.6e-95$, FDR $= 1.0e-92$) (Fig. 2).

## Clinical significance of miR-10b-5p in breast cancer

We analyzed the prognostic value of miR-10b-5p in four public databases: METABRIC ($n = 1262$), TCGA ($n = 1077$), GSE19783 ($n = 101$) and GSE40267 ($n = 181$). Kaplan–Meier survival analysis indicated significantly reduced overall survival in breast cancer patients with low miR-10b-5p expression in METABRIC database (HR $= 0.64$, 95% CI [0.52–0.8], $P = 4.1e-5$) (Fig. 3D). However, no significant prognostic effect of miR-10b-5p for breast cancer was found in either TCGA or GSE19783 (Fig. 3E and Fig. 3F). Noticeably, low expression of miR-10b-5p was correlated to higher overall survival (HR $= 1.92$, 95% CI [1.06–3.47], $P = 0.029$) in the GSE40267 dataset which is composed primarily of triple-negative (ER/PGR/HER2-negative) tumors (Fig. 3G).

According to the results from LinkedOmics, expression of miR-10b-5p was significantly related to PAM50 subtypes ($P = 3.934e-23$), ER. status ($P = 3.982e-2$), histological
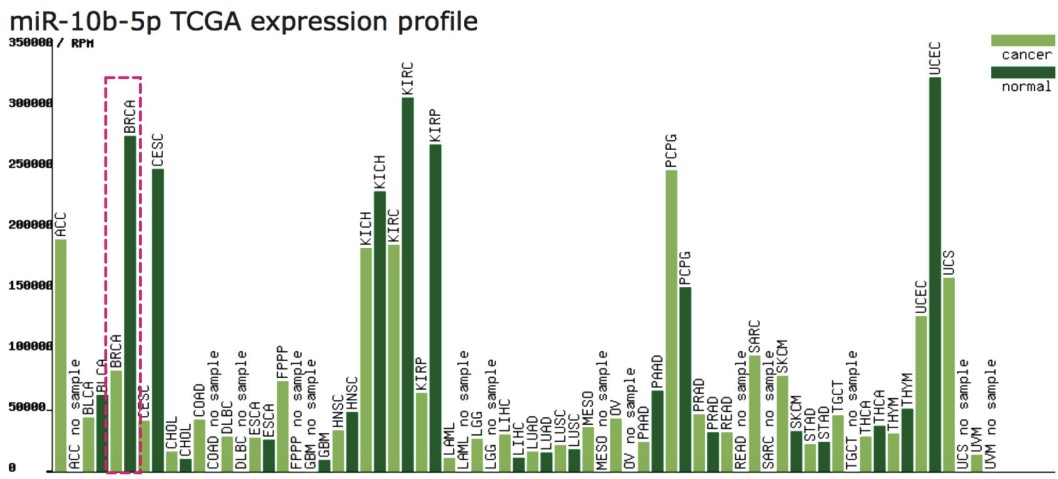

**Figure 1** **Expression profile of miR-10b-5p from TCGA.** The miR-10b-5p expression profile of various human cancer types was obtained from a TCGA data online analysis tool (http://bioinfo.life.hust.edu.cn/miR_path/index.html). MiR-10b-5p was down-regulated in breast adenocarcinoma tissues compared with normal tissues.

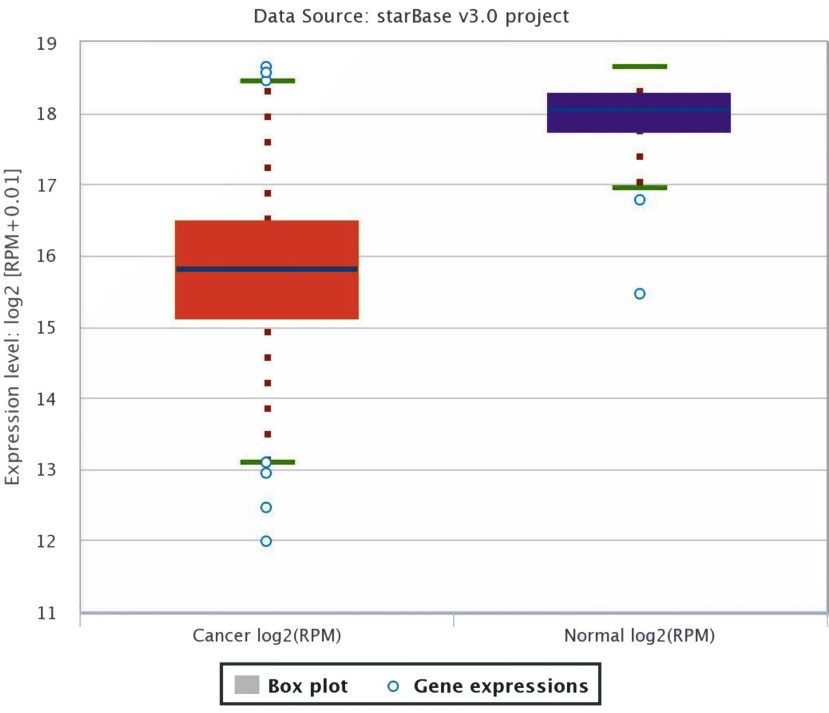

**Figure 2** **Expression level of miR-10b-5p from starBase v3.0 database.** The box plot was based on 1,085 breast cancer samples and 104 normal samples that revealed downregulated expression of miR-10b-5p in breast cancer.

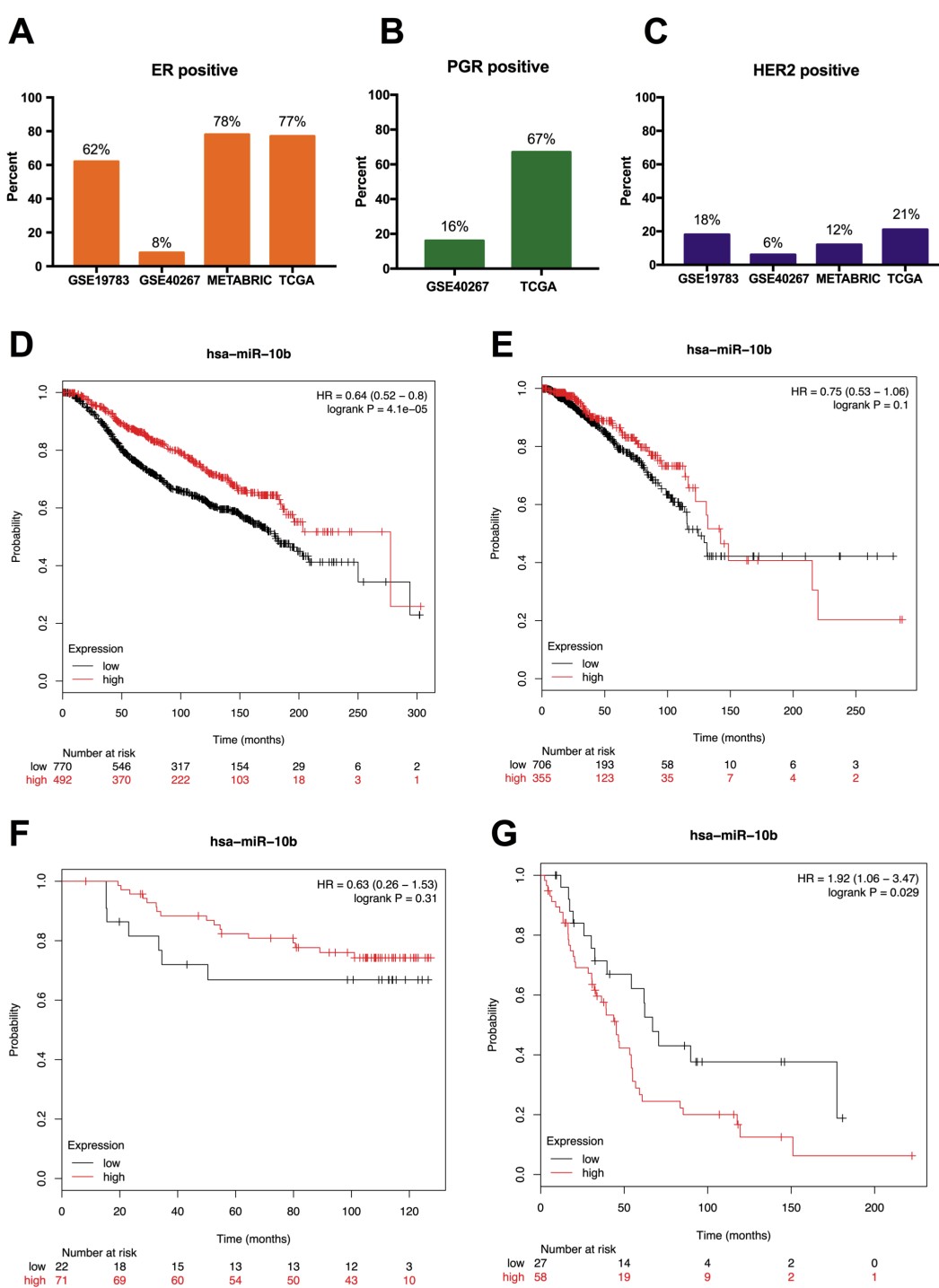

**Figure 3** **Prognostic value of miR-10b-5p in breast cancer from the KM plotter database.** ER, PGR, and HER2 status in four public databases (METABRIC, TCGA, GSE19783 and GSE40267) (A–C). Kaplan-Meier survival curves comparing patient survival time between samples with high- or low- level of miR-10b-5p expression in breast cancer in METABRIC (D), TCGA (E), GSE19783 (F), and GSE40267 database (G).

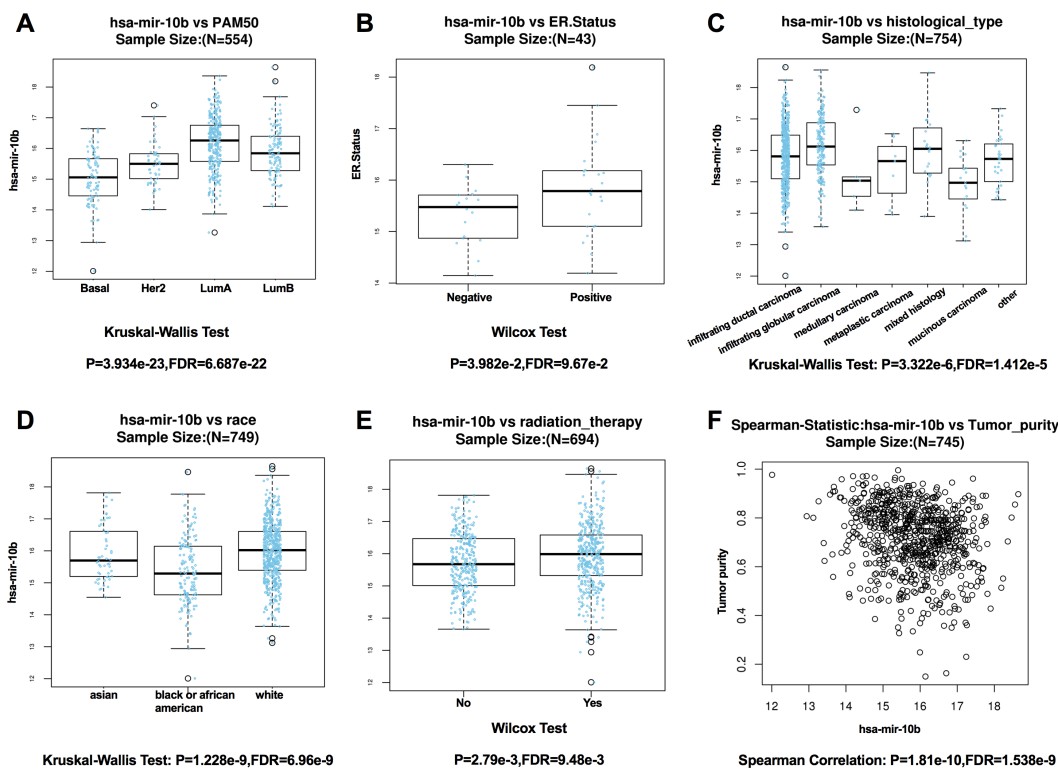

**Figure 4   Correlation between miR-10b-5p and clinicopathological features in breast cancerpatients from LinkedOmics database.** Box plot showing the relationship between miR-10b-5p expression and PAM50 subtypes (A) ER. status, (B) histological type, (C) patient race, (D) radiation therapy, (E) and tumor purity (F) in breast cancer.

type ($P = 3.322e-6$), patient race ($P = 1.228e-9$), radiation therapy ($P = 2.79e-3$), and tumor purity ($P = 1.81e-10$) (Fig. 4). However, no significant difference in miR-10b-5p expression was observed when patients were stratified by pathologic TNM stage, PR. status, and HER2.status (Table S1).

## Potential key targets of miR-10b-5p in breast cancer
Positively and negatively correlated genes of miR-10b-5p in breast cancer were presented as volcano plot (Fig. 5A). Fifty significantly positively correlated genes obtained from LinkedOmics were demonstrated in Fig. S1 and 48 significantly negatively correlated genes obtained from LinkedOmics were demonstrated in Fig. 5B. A total of 1,222 target genes of miR-10b-5p were predicted, and five overlapped genes, BIRC5 (Baculoviral IAP Repeat Containing 5), E2F2 (E2F Transcription Factor 2), KIF2C (Kinesin Family Member 2C), FOXM1 (Forkhead Box M1), and MCM5 (Minichromosome Maintenance Complex Component 5), were considered as candidate target genes of miR-10b-5p for further analysis (Fig. 5C). Inverse correlation and alignment of binding sites between miR-10b-5p and target genes in breast cancer were illustrated in supplemental Table S2 and Fig. S2.

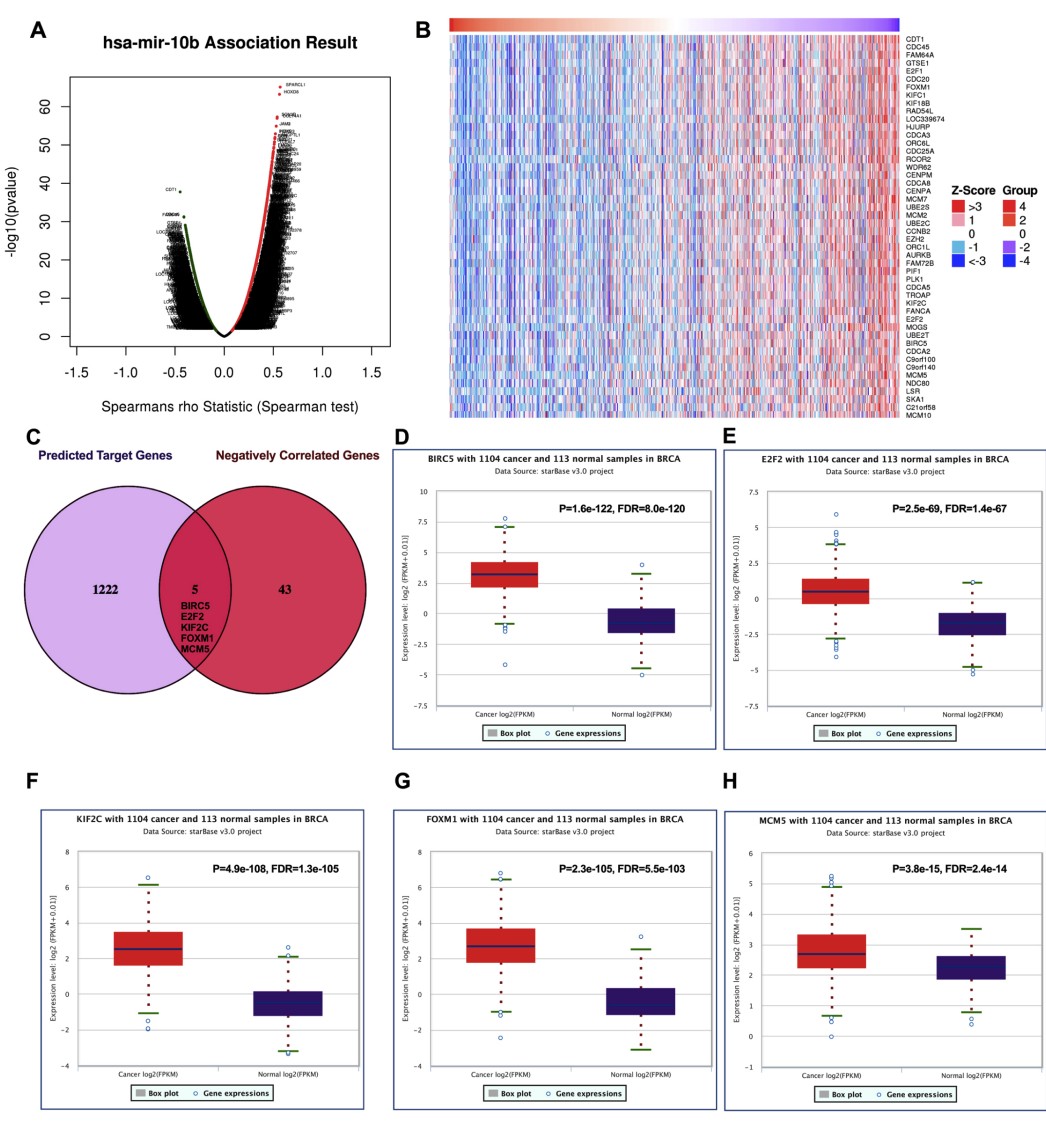

**Figure 5** **Predicting the potential target genes of miR-10b-5p in breast cancer.** The volcano plot showing the Log2 (fold change) vs. −log10 (*p*-value) obtained from LinkedOmics. The red dots represent positively correlated genes, and green dots represent negatively correlated genes of miR-10b-5p in breast cancer (A). A total of 48 significantly negatively correlated genes were acquired from LinkedOmics (B). A venn diagram showing the overlap between the target genes of miR-10b-5p predicted by starBase and negatively correlated genes from LinkedOmics. Five genes, BIRC5, E2F2, KIF2C, FOXM1, and MCM5, were considered as potential key target genes of miR-10b-5p (C). Expression levels of the potential target genes in breast cancer. Boxplots were based on 1104 breast cancer samples and 113 normal samples that revealed overexpression of target genes of miR-10b-5p in breast cancer (D–H).

We further investigated the expression levels of these five key target genes in breast cancer, and results showed that they were significantly up-regulated in cancer tissue samples (Figs. 5D–5H). Consistently, BIRC5, E2F2, FOXM, and MCM5, showed higher expression in breast cancer tissues than in normal tissues, according to the immunohistochemical

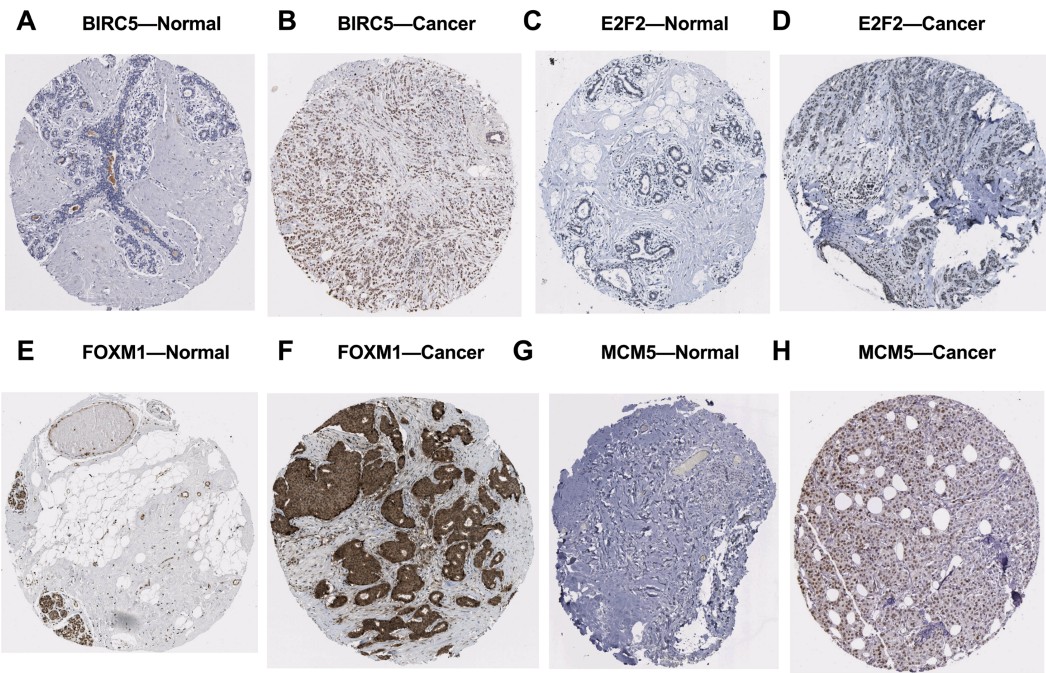

**Figure 6 Validation of the potential target genes of miR-10b-5p in breast cancer from the HPA database.** The Human Protein Atlas (HPA) database can be accessed at: https://www.proteinatlas.org. Expression levels of BIRC5, E2F2, FOXM1, and MCM5 were higher in breast cancer samples (B, D, F, H) than in normal samples (A, C, E, G).

assessment from HPA database (Fig. 6). However, data on KIF2C expression was not found in HPA.

## Functional analysis of target genes in cancer development

Analysis from CHAT revealed that BIRC5, E2F2, KIF2C, FOXM1, and MCM5 were mainly involved in sustaining proliferative signaling in cancer development, with npmi values of 0.15, 0.222, 0.168, 0.218, and 0.157, respectively. As illustrated, these target genes also play a critical role in evading growth suppressors, resisting cell death, and promoting genome instability and mutation during cancer progression (Fig. 7).

## Network construction and pathway enrichment analysis

In the PPI network, 5 nodes and 9 lines illustrated strong interactions (average node degree = 3.6, enrichment $P = 8.62e-09$) between the potential key target genes (Fig. 8). We further performed GO analysis and results revealed that these targeted genes were significantly enriched in biological processes (BP) of cell cycle, mitotic cell cycle process, mitotic cell cycle phase transition, establishment of chromosome localization, regulation of chromosome segregation, mitotic nuclear division, signal transduction by p53 class mediator, G2/M transition of mitotic cell cycle, chromosome segregation, and regulation of cell cycle (Table 1). The significant GO molecular function (MF) terms included sequence-specific double-stranded DNA binding and microtubule binding (Table 1). Cellular component (CC) enrichment displayed that genes were significantly present

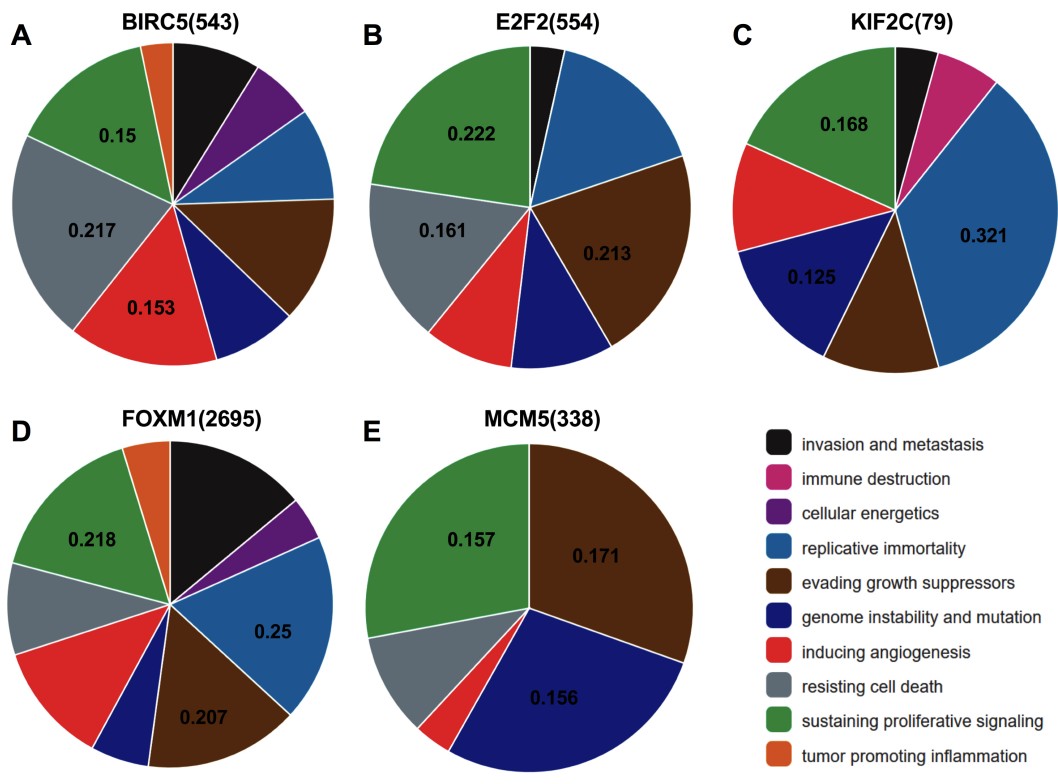

**Figure 7  Association between the target genes of miR-10b-5p and hallmarks of cancer from Cancer Hallmarks Analytics Tool (CHAT).** CHAT can be accessed at: http://chat.lionproject.net.

in microtubule associated complex, condensed chromosome kinetochore, microtubule, nucleoplasm, and nucleus (Table 1). In addition, using KEGG pathway analysis, target genes were found to be significantly involved in cellular senescence and cell cycle (Table 2). According to Reactome pathway analysis, these target genes were significantly enriched in cell cycle, assembly of the pre-replicative complex, mitotic G1-G1/S phases, and cell cycle checkpoints (Table 2).

## Measurement of target genes expression in MDA-MB-231 cells

As shown in Fig. 9, the expression levels of BIRC5 (A), E2F2 (B), FOXM1 (D) and MCM5 (E) were significantly up-regulated in MDA-MB-231 cells transfected with miR-10b antisense inhibitors compared with negative control (NC). A trend of increase was found for KIF2C (C) expression after transfection with inhibitors while there was no statistical significance. Moreover, MCM5 (E) was significantly down-regulated in cells transfected with miR-10b mimic as compared to NC.

## DISCUSSION

Despite advances in detection and therapies, breast cancer is still the leading cause of cancer related death in women (*Iorio et al., 2011*). MiR-10b has been implicated in regulating several human cancers, including breast cancer. However, the expression profile
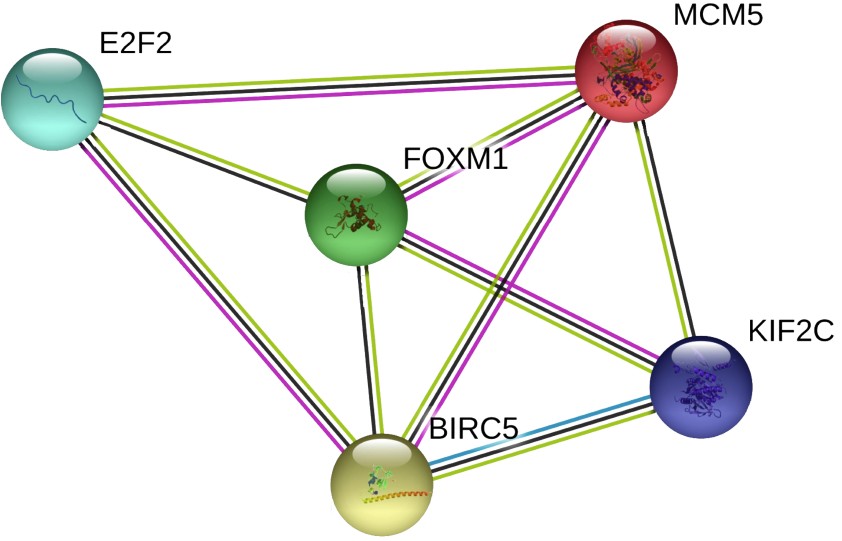

**Figure 8** **The PPI network of the target genes of miR-10b-5p in breast cancer.** The protein–protein interaction (PPI) network analysis was conducted using STRING v11.0. A total of 5 nodes and 9 edges constitute the network. Colored nodes denote query proteins. Lines with different colors represent the protein–protein interactions (blue: from curated databases; pink: experimentally determined; yellow: text-mining; black: co-expression).

**Table 1** **Functional annotation of the gene ontology (GO) terms.**

| GO-term | Description | Count in gene set | FDR |
|---|---|---|---|
| BP-GO:0007049 | Cell cycle | 5 of 1263 | 0.00041 |
| BP-GO:1903047 | Mitotic cell cycle process | 4 of 564 | 0.00062 |
| BP-GO:0044772 | Mitotic cell cycle phase transition | 3 of 254 | 0.0019 |
| BP-GO:0051303 | Establishment of chromosome localization | 2 of 72 | 0.0073 |
| BP-GO:0051983 | Regulation of chromosome segregation | 2 of 97 | 0.0101 |
| BP-GO:0140014 | Mitotic nuclear division | 2 of 136 | 0.0145 |
| BP-GO:0072331 | Signal transduction by p53 class mediator | 2 of 128 | 0.0145 |
| BP-GO:0000086 | G2/M transition of mitotic cell cycle | 2 of 123 | 0.0145 |
| BP-GO:0007059 | Chromosome segregation | 2 of 253 | 0.0428 |
| BP-GO:0051726 | Regulation of cell cycle | 3 of 1129 | 0.0429 |
| MF-GO:1990837 | Sequence-specific double-stranded DNA binding | 4 of 747 | 0.00088 |
| MF-GO:0008017 | Microtubule binding | 2 of 253 | 0.0277 |
| CC-GO:0005875 | Microtubule associated complex | 2 of 144 | 0.0164 |
| CC-GO:0000777 | Condensed chromosome kinetochore | 2 of 104 | 0.0164 |
| CC-GO:0005874 | Microtubule | 2 of 385 | 0.0267 |
| CC-GO:0005654 | Nucleoplasm | 4 of 3446 | 0.0267 |
| CC-GO:0005634 | Nucleus | 5 of 6892 | 0.0310 |

**Notes.**
BP, biological process; MF, molecular function; CC, cellular component; FDR, False Discovery Rate.
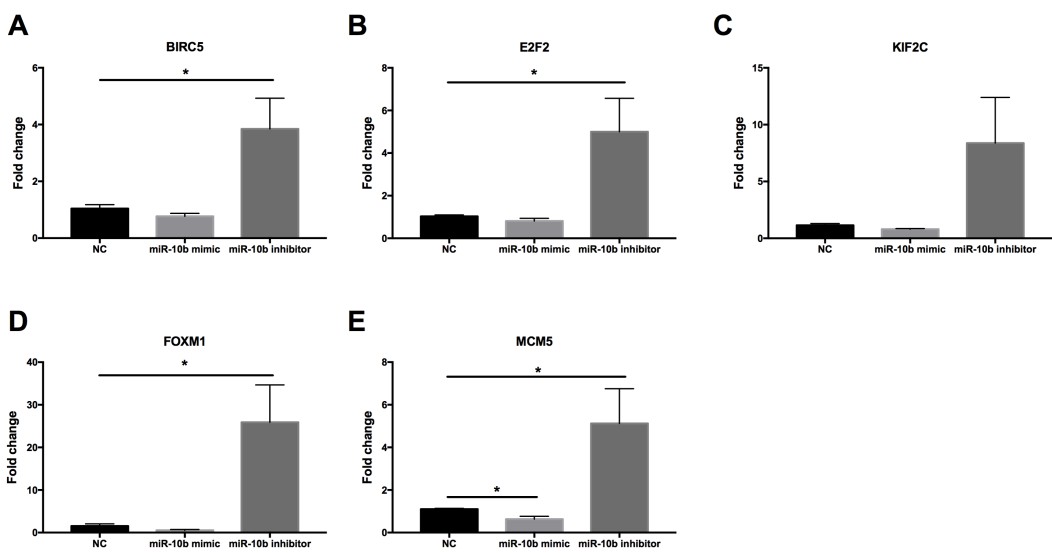

**Figure 9** **Quantitative RT-PCR analysis of target genes in MDA-MB-231 cells.** MDA-MB-231 cells were transfected with negative control (NC) or miR-10b mimic (50 nM) or miR-10b antisense inhibitors (100 nM). After 24 h of incubation, the expression levels of BIRC5 (A), E2F2 (B), KIF2C (C), FOXM1 (D) and MCM5 (E) were measured by quantitative RT-PCR analysis. * indicates $P < 0.05$.

**Table 2** **Functional annotation of the KEGG and Reactome pathway analysis.**

| Pathway | Description | Count in gene set | FDR |
|---|---|---|---|
| KEGG-hsa04218 | Cellular senescence | 2 of 156 | 0.0084 |
| KEGG-hsa04110 | Cell cycle | 2 of 123 | 0.0084 |
| Reactome-HSA-69278 | Cell Cycle, Mitotic | 4 of 483 | 9.84e−05 |
| Reactome-HSA-68867 | Assembly of the pre-replicative complex | 2 of 65 | 0.0020 |
| Reactome-HSA-453279 | Mitotic G1-G1/S phases | 2 of 145 | 0.0049 |
| Reactome-HSA-69620 | Cell Cycle Checkpoints | 2 of 265 | 0.0137 |

of miR-10b has led to conflicting reports and its role in tumorigenic process remains unexplored. In this work, we observed that miR-10b-5p expression level was significantly down-regulated in human breast cancer tissues than in adjacent normal tissues. These results were consistent with the work of *Iorio et al. (2005)* and *Fassan et al. (2009)*, who demonstrated that miR-10b was significantly down-regulated in breast carcinoma. *Ma, Teruya-Feldstein & Weinberg (2007)* reported that miR-10b was upregulated in metastatic breast cancer cell lines compared with primary mammary epithelial cells. However, their patient group was small and data on clinical variables limited.

After verification of miR-10b-5p expression in breast cancer, we investigated the associated prognostic value of miR-10b-5p in public databases. Our results indicated that low expression of miR-10b was significantly correlated to worse outcome in METABRIC database, while no prognostic significance of miR-10b-5p in breast cancer was found in either TCGA or the GSE19783 dataset. Interestingly, low expression of miR-10b-5p was associated with better prognosis in the GSE40267 database composed primarily

of triple-negative tumors. These inconsistent overall survival findings may be partially explained by differences in sample size, tumor subtypes, genetic status, diseases status, treatment received, economic and social status, etc.

Among the clinical characteristics evaluated, miR-10b-5p was significantly related to PAM50 subtypes, ER. status, histological type, patient race, radiation therapy, and tumor purity. However, no significant differences were observed based on pathologic TNM stage, PR. status, and HER2. status. In a recent study, miR-10b expression was observed to be inversely correlated with malignancy in human breast cancer (*Zhang et al., 2018*). However, *Gee et al. (2008)* found no significant association between miR-10b levels and metastasis or prognosis in a total of 219 patients with early breast cancer, while its expression correlated inversely with tumor size and grade. *Ma, Teruya-Feldstein & Weinberg (2007)* and *Ma et al. (2010)* showed a positive correlation between miR-10b expression and cell migration and invasion, and silencing of miR-10b could inhibit metastasis in a mouse cancer model. Others have reported down-regulated miR-10b levels in breast cancer and indicated that restoration of miR-10b expression might have therapeutic value (*Fassan et al., 2009*; *Andorfer et al., 2011*). Additionally, *Biagioni et al. (2012)* and *Biagioni et al. (2013)* characterized miR-10b* (miR-10b-3p) as a tumor suppressor microRNA in primary breast cancers and the locus of microRNA-10b was a critical target for breast cancer insurgence and dissemination.

To facilitate a more in-depth understanding of the role of miR-10b-5p in breast cancer, we analyzed the interaction networks between the key target genes of miR-10b-5p and their potential functions in breast cancer. Although 1222 target genes of miR-10b-5p were predicted and 48 significantly negatively correlated genes were obtained, only 5 overlapped genes, BIRC5, E2F2, KIF2C, FOXM1, and MCM5, were considered as potential key targets of miR-10b-5p in breast cancer for further analysis. Results demonstrated that these 5 key target genes were significantly up-regulated in cancer tissue samples. According to the immunohistochemical assessment from HPA database, BIRC5, E2F2, FOXM, and MCM5, exhibited higher expression in breast cancer tissues than in normal tissues while data on KIF2C expression was not found. Consistently, our qRT-PCR analysis also showed that expression levels of BIRC5, E2F2, FOXM1 and MCM5 were significantly up-regulated in human breast cancer cell line MDA-MB-231 transfected with miR-10b antisense inhibitors and a trend of increase was also found for KIF2C expression, further validating the predicted target genes of miR-10b-5p in the current study. BIRC5, is a member of the inhibitor of apoptosis (IAP) gene family, which encode negative regulatory protein (also known as survivin) that functions as a key regulator of mitosis and programmed cell death (*Mita et al., 2008*). BIRC5 has been reported to be selectively expressed in most tumors, including breast carcinomas, and yet low in adult tissues (*Tanaka et al., 2000*). E2F2, a member of the E2F family of transcription factors, is known to play a key role in the control of cell cycle progression and proliferation (*Ren et al., 2002*). *Hollern et al. (2014)* observed striking reductions in metastatic capacity and in the number of circulating tumor cells in E2F2 knockout mice using a murine model of breast cancer, suggesting a crucial role for E2F2 in tumor development and metastasis. KIF2C encodes a kinesin-like protein that functions as a microtubule-dependent molecular motor and a key regulator of mitotic

spindle assembly and dynamics (*Gwon, Cho & Kim, 2012*). *Shimo et al. (2008)* confirmed KIF2C overexpression in breast cancer cells and its phosphorylation in G(2)/M phase. Their findings indicate that overexpression of KIF2C might be involved in breast carcinogenesis and be a potential therapeutic target for breast cancers. FOXM1 is a transcriptional activator involved in cell proliferation, which stimulates cell cycle progression and inhibits apoptosis (*Wierstra, 2013*). It has been considered as a key gene that serves important roles in multiple biological processes in triple-negative breast cancer and a promising potential target for the prevention and/or therapeutic intervention in cancer treatment (*Tan et al., 2019*). Previous research showed a strong expression of FOXM1 in clinical tissues of human breast cancer, and knockdown of FOXM1 expression diminished the proliferation and anchorage-independent growth of breast cancer cells (*Yang et al., 2013*). MCM5 is a member of the MCM family of chromatin-binding proteins and actively participates in cell cycle regulation. *Snyder, He & Zhang (2005)* demonstrated that, in addition to its roles in DNA replication, MCM5 was also necessary for transcription activation. Expression profiling of MCM5 in multiple malignancies has been reported (*Giaginis et al., 2011*; *Das et al., 2013*; *Eissa et al., 2015*). MCM5 was shown to be significantly over-expressed in cervical cancer and clinically correlated to cervical carcinogenesis, implying that it may serve as potential diagnostic and prognostic marker for human malignancies (*Das et al., 2013*). *Eissa et al. (2015)* found that MCM5 expression was positive in breast cancer patients and high levels of MCM5 were associated with short relapse free survival of breast cancer. They also identified MCM5 expression changes consistent with the miRNA-10b target regulation, and considered both miR-10b and MCM5 as prognostic biomarkers in breast cancer. In line with previous studies, our CHAT analysis revealed that these potential key target genes were mainly involved in sustaining proliferative signaling, evading growth suppressors, resisting cell death, and promoting genome instability and mutation in cancer development.

In our study, the PPI network analysis illustrated strong interactions between the potential key target genes. In a previous study, *Zhao et al. (2014)* identified KIF2C as a novel FOXM1 transcriptional target that may be implicated in the acquisition of chemoresistance in cancer treatment. Another study conducted by *De Moraes et al. (2015)* showed that FOXM1 could target BIRC5 to modulate breast cancer survival and chemoresistance. *Sullivan et al. (2012)* found that FOXM1 could also regulate E2F2 transcription, as evidenced by the fact that the transcription levels of E2F2 was significantly decreased with the knockdown of FOXM1. Additionally, a previous study reported that E2F2 intercalated in the Rb pathway bound to discrete sites in the BIRC5 promoter, and repressed its transcription (*Guha & Altieri, 2009*). These findings of previous studies, thus, confirm our results from the PPI network analysis.

According to the GO analysis, the target genes of miR-10b-5p were significantly enriched in cell cycle, mitotic cell cycle process and phase transition, establishment of chromosome localization, regulation of chromosome segregation, mitotic nuclear division, signal transduction by p53 class mediator, G2/M transition of mitotic cell cycle, chromosome segregation, sequence-specific double-stranded DNA binding, and microtubule binding, suggesting that miR-10b-5p might impact the development of breast cancer by participating

in the biological processes and molecular functions mentioned above. In addition, the KEGG and Reactome pathway analysis revealed that these target genes were significantly involved in cell cycle, cellular senescence, assembly of the pre-replicative complex, mitotic G1-G1/S phases, and cell cycle checkpoints. Thus, these data suggest that miR-10b-5p functions to impede breast carcinoma progression by regulating the above-described pathways.

## CONCLUSION

In conclusion, miR-10b-5p is down-regulated in breast cancer and might serve as a potential diagnostic and prognostic marker for breast cancer. The tumor-suppressing effect of miR-10b-5p might be mediated via regulation of key target genes involved in cell cycle. In vivo or *in vitro* experiments are warranted to verify the underlying mechanism of miR-10b-5p and its interactions with target genes in the future.

## ACKNOWLEDGEMENTS

The results shown here are in whole or part based upon data generated by the TCGA Research Network: https://www.cancer.gov/tcga.

### Funding

This research was supported by the National Natural Science Foundation of China (No. 31802075). There was no additional external funding received for this study. The funders had no role in study design, data collection and analysis, decision to publish, or preparation of the manuscript.

### Grant Disclosures

The following grant information was disclosed by the authors:
National Natural Science Foundation of China: 31802075.

### Competing Interests

The authors declare there are no competing interests.

### Author Contributions

- Junmin Wang performed the experiments, analyzed the data, prepared figures and/or tables, approved the final draft.
- Yanyun Yan and Zhiqi Zhang analyzed the data, prepared figures and/or tables, approved the final draft.
- Yali Li conceived and designed the experiments, contributed reagents/materials/analysis tools, authored or reviewed drafts of the paper, approved the final draft.

## Data Availability

qRT-PCR data is available in the Supplemental Files.

miR-10b-5p expression profile of various human cancer types is available at a TCGA data online analysis tool: http://bioinfo.life.hust.edu.cn/miR_path/.

Expression level of miR-10b-5p (MIMAT0000254) in breast cancer is available at starBase v3.0 project: http://starbase.sysu.edu.cn/panMirDiffExp.php.

Prognostic value of miR-10b-5p in breast cancer is available at the Kaplan-Meier Plotter Database (KMPD): http://kmplot.com/analysis/index.php?p=service&cancer=breast_mirna.

Relationship between miR-10b-5p and clinical features is available at LinkedOmics: http://linkedomics.org.

Immunohistochemical assessment of BIRC5, E2F2, FOXM1, and MCM5 is available at the Human Protein Atlas (HPA) database v18.1:

https://www.proteinatlas.org/ENSG00000089685-BIRC5/pathology/tissue/breast+cancer#img.

https://www.proteinatlas.org/ENSG00000007968-E2F2/pathology/tissue/breast+cancer#img.

https://www.proteinatlas.org/ENSG00000111206-FOXM1/pathology/tissue/breast+cancer#img.

https://www.proteinatlas.org/ENSG00000100297-MCM5/pathology/tissue/breast+cancer#img.

CHAT can be accessed using queries BIRC5, E2F2, KIF2C, FOXM1, and MCM5 at: http://chat.lionproject.net/?q=kif2c&q=mcm5&q=foxm1&q=e2f2&q=birc5&measure=npmi&chart_type=pie&hallmarks=top.

This is a secondary analysis of a public dataset.

## Supplemental Information

Supplemental information for this article can be found online at http://dx.doi.org/10.7717/peerj.7728#supplemental-information.

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
