# Peer review of "Role of miR-10b-5p in the prognosis of breast cancer"

_PeerJ, doi:10.7717/peerj.7728_

## Round 0.1 · original submission · Major Revisions

Our reviewers have thoroughly gone through your manuscript and they recommend major suggestions that need to be addressed. Please provide justification for the manuscript title, as requested by the Reviewer 1. Address all the concerns raised by both the reviewers. Pay attention to the English grammar and if possible get it assessed for language.

·

Basic reporting

In general english is clear and background literate is well-cited. Authors should provide more details in figure legends.

Experimental design

Overall computational analysis is well structured however it lacks the experimental validation.

Validity of the findings

Experimental validation needs more strength.

Additional comments

In this manuscript authors report down-regulation of miR-10b-5p in breast cancers and its role in tumorigenesis. However, I failed to see a single experiment which describe the role of miR-10b-5p in 'tumorigenesis'. Tumorigenesis means initiation/formation of tumor but none of the experiment or analysis address this in manuscript. I will suggest either authors change the title accordingly or provide evidence for 'Role of miR-10b-5p in the tumorigenesis of breast cancer'.

Specific points:
1) What is expression pattern of miR-10b-5p in Basal, Luminal, and Her2+ breast cancer cell lines?
2) Does knock-down of miR-10b-5p reduce the expression of BIRC5, E2F2, KIF2C, FOXM1 and MCM5?
3) Does miR-10b-5p down-regulation impacts tumor initiating property of breast cancer cell lines?
Minor point:
1) Fig.4 x and y axis font size should be bigger, difficult to read in current size.

Reviewer 2 ·

Basic reporting

The manuscript by Wang et al predicted that miR-10b-5p a micro RNA is downregulated in breast cancer cells and it targets five important genes which are involved in cell growth cycle. The level of these predicted target genes as expected upregulated in breast cancer cells. All these analyses were done by using various bioinformatic tools. The paper is well written, and conclusions are mostly supported by data. The paper is interesting. However, the novelty of this paper is not clear at all considering it is well known that miR-10b is involve in tumorigenesis (Li Ma, 2010 review breast cancer research).

Introduction; its not up-to-date. Please add references from recent articles as there are many interesting studies on breast cancer and miR-10b relationship.

Ref. Dai et al. does not have DOI.
None of the address has vol no, issue no, and page numbers which are critical. Please add them as per journal’s format.
Considering combining table 1,2, and 3 under three subheadings such as biological, cellular and molecular functions. Having extra tables with just two values does not improve the manuscript.
Although the manuscript is written in clear English, there are some sentences that need to be modified such as Line 31, 32, and 197 “ was found to be significantly 32 correlated to worse outcome”.
L 53 needs reference/citation
Fig3 ligand (A) label should go at the end of the sentence to maintain the consistency.
Fig5 ligand order of alphabet numbering are at the beginning which is different than other ligands.
Fig8 ligand needs more information. Full form of PPI should be written as Fig should stand by itself. The detail of 3 colored line should be elaborated.
More information required for some figure ligands. Such as figure 1 how many samples were considered? What was the basis of selection as some of the cancer cell line does not have even normal cells data? Authors may consider changing “Abnormal expression” in Figure 1 ligand.

Experimental design

Need to add more how the question identified the gap area as miR-10b was reported as possible cancer marker by multiple groups. Author can emphasis on the target search.

Validity of the findings

Figure 1. Thanks for providing this very simple but meaningful figure. However, there are some cancer cell data such as LAML, FPPP and LGG does not have their control (normal cells). In absence of control I don’t think these data are adding any value to the result.
Since reports on miR-10b expression in cancer cells are controversial in literature, please provide reason why expression of micro RNA is high in PCPG cancer cells than normal cells?
Figure 2. Why sample numbers for cancer cells are almost 10 time less than normal cells(control) were considered for box plot analysis?
Controversy on miR-10b level can be partially explained due to Heterogeneity of miR-10b expression in circulating tumor cells (Gasch et al, scientific reports 2015).
All the data are predictions, I would strongly suggest to at least check the expression of these 5 targes(BIRC5,E2F2, KIF2C, FOXM1, and MCM5) by either northern blot or RT-PCR under overexpression of micro RNA 10b in breast cancer cell line. This very simple experiment will give a direct evidence of regulation.

Since there are multiple reports on high level of miR-10b RNA which may be due to heterogeneous cell population but still deserve further analysis. positively correlated genes should also be analysed and added as a supplemental material. This knowledge will help researcher in the field.

---

## Round 0.2 · Minor Revisions

Dr. Li,

The reviewers feel that you have addressed most of the concerns satisfactorily. However it is necessary to incorporate a brief discussion on RT-PCR as recommended by Reviewer 2 before I can accept the manuscript.

·

Basic reporting

No comment

Experimental design

No comment

Validity of the findings

No comment

Additional comments

Authors and carefully address all the concerns raised in previous version.

Reviewer 2 ·

Basic reporting

No comment

Experimental design

No comment

Validity of the findings

New RT PCR results should be discussed under the discussion section showing the bioinformatical analysis prediction is Indeed true.

Additional comments

The manuscript has improved but still, need some corrections such as the result of RT-PCR has not been discussed. Author needs to write both how these RT-PCR result helped in overall conclusion and compare these result with the literature of other target genes reported.

---

## Round 0.3 · Minor Revisions

Hi Dr. Li,
Before I make the final decision can you check if you need to add the word 'antisense' after miR-10b on line 250? If yes please make that change and re-submit.
Thanks

---

## Round 0.4 · accepted · Accept

Dr Li,

Congratulations, your manuscript has been accepted for publication.